# Utility of FDG PET/CT in Patient with Synchronous Breast and Colon Cancer

**DOI:** 10.3390/diagnostics13132293

**Published:** 2023-07-06

**Authors:** I-Lin Su, Yen-Kung Chen

**Affiliations:** 1Department of Nuclear Medicine and PET Center, Shin Kong Wu Ho-Su Memorial Hospital, Taipei 11101, Taiwan; ellen00418@gmail.com; 2School of Medicine, Fu Jen Catholic University, New Taipei City 24205, Taiwan

**Keywords:** FDG PET/CT, synchronous tumor, breast cancer, colon cancer

## Abstract

The most common malignancy in women is breast cancer, and the second one is colon cancer. Synchronous breast and colon cancers are rare. Here, we reported a 60-year-old woman with a left breast mass for six months. Biopsy revealed an invasive ductal carcinoma. She underwent 2-[Fluorine-18]fluoro-2-deoxy-D-glucose (FDG) positron emission tomography (PET)/computed tomography (CT) scan for evaluation of the extent of the disease. FDG PET/CT revealed an advanced left breast cancer with multiple metastases in both regional and distant lymph nodes (in left axilla level I/II, lower paratracheal region, and right lung hilum), bilateral lungs, and axial and proximal appendicular skeletons. An early staged synchronous colon cancer was detected incidentally on FDG PET/CT images. After endoscopic mucosal resection of colon cancer, she received palliative chemotherapy for breast cancer with a marked therapeutic response. The disease status of post-treated breast cancer remained relatively stationary for more than one year. Brain metastasis was noted afterward. Nevertheless, there was no evidence of colon cancer recurrence throughout her breast cancer disease course.

## 1. Introduction

Breast cancer is the most frequent cancer and the leading cause of cancer death among females worldwide [1]. In Taiwan, breast cancer is also the most common cancer among women, and the incidence is rising [2]. In Asia, reports have indicated that the annual incidence of breast cancer has doubled or tripled over the past two decades [3].

Colorectal cancer is a major healthcare problem, being one of the most significant causes of cancer death worldwide and imposing an enormous global burden as a result of aging and the expansion of populations in both developing and developed countries. There is evidence that screening can reduce colorectal cancer mortality by identifying cancer at an early stage and through the removal of clinically significant colorectal adenomas. However, one controversial issue is what screening tool should be used. Although most evidence points toward fecal occult blood testing as a feasible method for screening colorectal cancer, the sensitivity of a one-time test ranges from 30–60% [4,5]. The other problem of fecal occult blood testing is its high false positivity, which may lead examinees to receive unnecessary investigations such as colonoscopies.

Synchronous tumors are defined as distinct tumors arising simultaneously or diagnosed in a range of less than 6 months apart. The incidence of multiple primary malignancies is 0.52–11.7%, and it is believed that the incidence has increased due to increased life expectancy and progressive advances in diagnostic techniques [6]. A cancer survey aims to detect cancer at an early stage when it is treatable and curable. It is recognized that 2-[Fluorine-18]fluoro-2-deoxy-D-glucose (FDG)- positron emission tomography (PET)/computed tomography (CT) is useful in managing various cancer types because glucose metabolism is generally activated in malignancy. We present a case in which synchronous early-stage colon cancer was detected incidentally on an FDG PET/CT scan for the staging of breast cancer.

## 2. Case Report

A 60-year-old woman had a family history of breast cancer. Her sister was diagnosed with breast cancer younger than age 40. Her mother was diagnosed with breast cancer over the age of 70. She presented with a left breast mass with intermittent tenderness for six months. Left shoulder pain and mild weakness in left arm elevation for several months were also complained. On physical examination, a tumor was palpated in the left breast at 5 o’clock position. Sonography revealed an irregular-shaped 4.4 × 3.1 cm tumor in her left breast with heterogeneous internal echo, hypervascularity around the tumor, and posterior acoustic shadowing. The tumor was given a Breast Imaging-Reporting and Data System score of 4a (BIRADs 4a). Core needle biopsy revealed an invasive ductal carcinoma with estrogen receptor (ER) negative, progesterone receptor (PR) negative, and human epidermal growth factor receptor (HER2) 1+ score. The proliferative index (ki67) of tumor cells is 90%. On laboratory investigations, the patient’s serum carcinoembryonic antigen (CEA) and cancer antigen 15-3 (CA15-3) were in the normal ranges. As a part of the pre-operative cancer staging, an FDG PET/CT scan (Simens, Biograph mCT, Munich, Germany) was performed after 8 h of fasting. FDG PET/CT scan showed a central necrotic tumor measuring 5.9 cm with a maximum standard uptake value (maxSUV) of 23.3 in the left breast, extending to the skin, which corresponded to the left breast tumor seen on sonography. Several smaller left breast nodules with various degrees of FDG uptake are found. Furthermore, bilateral lung nodules, bony lesions in axial and proximal appendicular skeletons (skull, facial bones, cervical spine, thoracic spine, lumbar spine, sacrum, bilateral scapulae, bilateral clavicles, bilateral humeri, bilateral ribs, sternum, bilateral pelvis, and bilateral femora), and multiple lymph nodes in the left axilla, lower paratracheal space, and right lung hilum with mild to intense FDG uptake are also noted. Advanced left breast cancer with multiple regional nodal, distant nodal, pulmonary, and bony metastases was depicted and classified as clinical T4bN2M1, stage IV, according to the American Joint Committee on Cancer (AJCC), 8th edition. Aside from the aforementioned findings, focal intense FDG uptake in the sigmoid colon was found (Figure 1). A delayed scan with per-rectal administration of laxative-augmented contrast medium (Fleet enema, Lynchburg; Xenetix, Gurbet) revealed a 1.5 cm nodular filling defect with a maxSUV of 50.8 in the sigmoid colon. Subsequent colonoscopy was performed smoothly with good bowel preparation and reached the cecum. A 1.5 cm polyp in the sigmoid colon was disclosed and removed with piecemeal endoscopic mucosal resection (Figure 2). Histopathology report of the specimen revealed colonic mucosa with the proliferation of atypical glandular cells arranged in irregular glands infiltrating in the desmoplastic stroma, suggesting an adenocarcinoma.

The patient was treated with palliative chemotherapy with paclitaxel plus cisplatin for breast cancer. In addition, palliative radiotherapy for local pain control (left breast, left scapula, sternum, and spine) was given. A second follow-up FDG PET/CT scan 3 months later during treatment revealed marked regression of prior breast cancer and multiple nodal, pulmonary, and bony metastases. A series of follow-up FDG PET/CT scans 8 months (Figure 3) and 11 months later revealed the stationary status of the post-treated breast cancer and metastatic lesions. However, the patient presented with progressive slurred speech, right arm weakness, and right leg weakness 17 months after the first FDG PET/CT scan. Contrast-enhanced magnetic resonance imaging of the brain demonstrated at least five tumors with nodular or rim-like enhancement and perifocal vasogenic edema in the right parietal lobe, right occipital lobe, left precentral gyrus, and right cerebellum, suggesting brain metastasis. Rapid progression of the disease was noted later on. Thus, the patient and her family decided to receive hospice care.

## 3. Discussion

Metastatic breast cancer may spread to any part of the body. At the time of diagnosis, approximately 5% to 10% of patients will harbor lymph nodes or distant metastases [7]. It most often spreads to the bones, liver, lungs, and brain. In our patient, she had left breast invasive ductal carcinoma with left axillary lymph nodes, multiple bones, and bilateral lung metastases. Breast cancer also has a tendency to metastasize to the gastrointestinal tract, with previous reports placing the stomach and small intestines among the most common sites. Colonic and rectal metastases occur less frequently or are both less recognized and diagnosed [8]. Colon metastasis typically presents as an invasion of the submucosal layer through blood vessels or the lymphatic system. Gastrointestinal metastasis from breast origin is rare in clinical practice. A study showed that patients with breast cancer have a 25% increased risk of developing a second cancer. Breast cancer and colon cancer are rarely observed synchronously. The incidence of breast and colon cancer in women detected at the same time is 3.85% [9]. The clinical and pathological features of synchronous tumors of the breast and colon are not fully established, and controversy exists as to the relationship between the two. There is a correlation between family history and synchronous tumors. A specific genetic mutation, cell cycle checkpoint kinase 2 1100delC (CHEK2 1100delC), has been described in patients with hereditary breast and colorectal cancer phenotypes [10]. CHEK2 1100delC is also found to be an important breast cancer-predisposing gene and is associated with shorter recurrence-free survival [11]. Our patient did not receive genetic sequencing.

FDG PET depicts the complete oncology application with a sensitivity and specificity of 84% and 88%, respectively [12]. Evidence exists that reductions in colorectal cancer mortality can be achieved through the detection and treatment of early-stage colorectal cancers [13]. Colonoscopy is the gold standard for the diagnosis of colonic neoplasms. Nevertheless, colonoscopy can be complicated by perforation, hemorrhage, and respiratory depression due to sedation, arrhythmia, transient abdominal pain, ileus, and nosocomial infection. The quality of bowel preparation could also have a certain degree of influence on the detection rate of colonoscopy. The FDG PET scan offers an alternative way of examining the entire colonic metabolism of glucose. FDG PET depicts the primary colorectal carcinomas with sensitivity from 94% to 100% [14,15]. Incidental FDG uptake is usually encountered in the colorectum. Conventionally, increased colorectal FDG uptake might be indicative of an underlying disease such as colorectal neoplasia. However, the subsequent colonoscopic evaluation does not reveal any clinically relevant colorectal neoplasia in a considerable number of examinees with incidental FDG uptake in the colorectum. Benign, infectious, inflammatory, and granulomatous processes may also cause an increase in FDG uptake. These hypermetabolic areas of the colorectum could be the cause of the false-positive findings confounding the reads. In fact, most of these false-positive findings might result from the physiologic FDG accumulation in the colorectum that has been postulated as colorectal wall uptake due to fecal irritation and/or intraluminal accumulation due to secretion into the lumen.

The use of a laxative-augmented contrast medium before a delayed FDG PET/CT scan leads to a reduction in the number of false-positive findings and increases the accuracy in the detection of colorectal cancer [16]. Delayed FDG PET/CT performed after administration of a laxative-augmented contrast medium might be useful for identifying patients needing additional diagnostic procedures or avoiding unnecessary colonoscopic evaluation. The rectosigmoid protocol consists of an initial phosphosoda enema (Fleet; C.B. Fleet, Lynchburg, Va). Following the evacuation, 500 mL of diluted 3% contrast medium was instilled into the anus. In our case, we used per-rectal administration of laxative-augmented contrast medium. An FDG-avid filling defect was discovered on FDG PET/CT scan. Like the PET/CT mapping technique, a laxative-augmented contrast medium may be considered a sort of functional and anatomic fusion application because the laxative has a functional effect to facilitate bowel motility, whereas the contrast agent allows anatomic representation to illustrate the bowel structures (Figure 1C’). In the previous study, we applied this imaging protocol to help decrease the false positive rate of colorectal FDG uptake from 14.4% (1464 participants) at initial PET to 0.9% (93 participants) at delayed PET/CT. The proportion of colon adenoma to adenocarcinoma detected by FDG PET/CT scan is about 73% to 27% [17]. Our patient received a colonoscopy one week later, and a 1.5 cm polyp in the sigmoid colon was found, which was proved to be adenocarcinoma.

FDG PET/CT is a valuable imaging tool in the detection and staging of multiple primary malignancies. One should always keep in mind that synchronous malignancies could happen occasionally. To differentiate a second malignancy of a different origin from metastasis, some strategies that could be used. First, recognizing the common metastatic location of the primary malignancy. An unusual metastatic lesion site should raise suspicion of another primary malignancy. Second, different types of malignancies may show different metabolic profiles and different responses to treatment. Thus, synchronous malignancies may present different FDG avidity. Discordant therapeutic response to chemotherapy, radiotherapy, etc., may also suggest synchronous malignancy sometimes.

The prognosis of synchronous malignancies is evaluated separately according to the stage of each malignancy. Our patient was diagnosed with imaging stage T4bN2M1 (stage IV) breast cancer and pathological stage T1 sigmoid colon cancer. Early detection and recognition of the second primary malignancy are crucial and may bring benefits to the patient. After the removal of the malignant colonic polyp via endoscopic mucosal resection, our patient received palliative chemotherapy with paclitaxel plus cisplatin for breast cancer. A follow-up FDG PET/CT scan 3 months later during treatment revealed marked regression of prior breast cancer and multiple nodal, pulmonary, and bony metastases. A series of follow-up FDG PET/CT scans 8 months and 11 months later revealed the stationary status of the post-treated breast cancer and metastatic lesions. Brain metastasis with rapid disease progression was noted 17 months after the first FDG PET/CT scan, consistent with the poor prognosis of triple-negative breast cancer. Nevertheless, no evidence of colon cancer recurrence was noted on FDG PET/CT scan throughout the patient’s breast cancer disease course.

Synchronous malignancies could occasionally happen, making the patient’s clinical scenario more complicated. Early detection of the second primary malignancy is important. Physicians and patients can put full effort into the management of the more advanced first-diagnosed cancer after dealing with the relatively early staged second primary malignancy. Treatment plans can also be made covering both malignancies at the same time. FDG PET/CT is a valuable imaging tool for detecting synchronous malignancies. Delayed FDG PET/CT with laxative-augmented contrast medium is useful in identifying colorectal lesions.

## Figures and Tables

**Figure 1 diagnostics-13-02293-f001:**
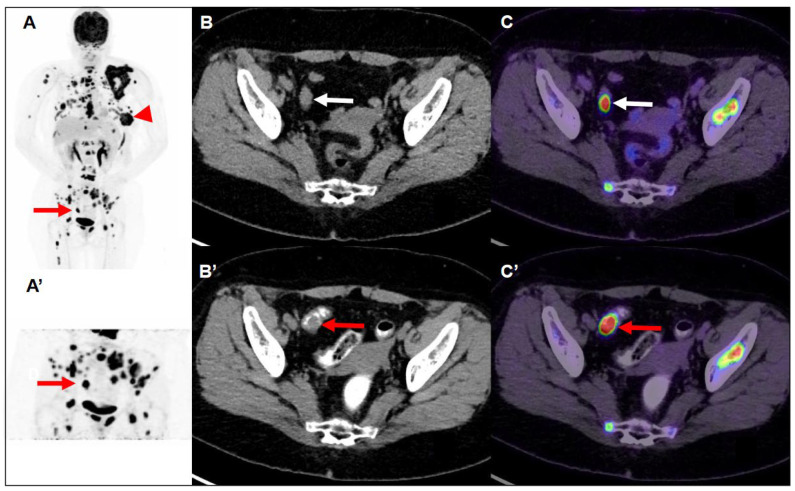
Maximum-intensity-projection image of FDG PET/CT (**A**) demonstrated multiple nodules with intense FDG uptake in the left breast (Largest: central necrosis, 5.9 cm, maxSUV:23.3) extending to the skin (arrowhead). There are lymph nodes with mild increased to intense FDG uptake in the left axilla level I/II and lower paratracheal and right hilar. There is focal intense FDG uptake in the skull, facial bone, C/T/L/S-spine, bilateral scapulae, bilateral clavicle, bilateral humerus, bilateral rib cages, sternum, bilateral pelvic, and bilateral femoral bone. Multiple nodules with intense FDG uptake in the bilateral lungs. So, it is advanced left breast cancer with multiple nodal, pulmonary, and bony metastases. In addition, transaxial CT (**B**) and fused FDG PET/CT images (**C**) demonstrated focal intense FDG uptake (maxSUV 60.9) in the sigmoid colon (white arrow), but the hypermetabolic focus could not be clearly delineated on the CT images. So, 40 min later, maximum-intensity-projection (**A’**), transaxial CT (**B’**), and fused PET/CT images (**C’**) of delayed FDG PET/CT scan after per-rectal administration of laxative-augmented contrast medium revealed a filling defect with persistent intense FDG uptake (maxSUV 50.8) in the sigmoid colon (red arrow), suggesting a hypermetabolic space-occupying lesion in bowel lumen. The FDG PET/CT images were acquired using a Siemens Biograph mCT (PET/CT) scanner (Siemens Medical Solutions) with an average spatial resolution of 4.4 mm at 1 cm and of 5.0 mm at 10 cm from the transverse field of view (FOV) and a maximum sensitivity of 8.1 kcps/MBq at the center of the FOV. After IV administration of 185 MBq (5 mCi) of FDG and a standard uptake period of 60 min, emission images were acquired for 3 min per bed position. Images were reconstructed using proprietary Siemens HD PET software (XP) with the iterative TrueX + TOF OSEM method. In addition, the images were analyzed semi-quantitatively using the standard uptake value (SUV) in the patient.

**Figure 2 diagnostics-13-02293-f002:**
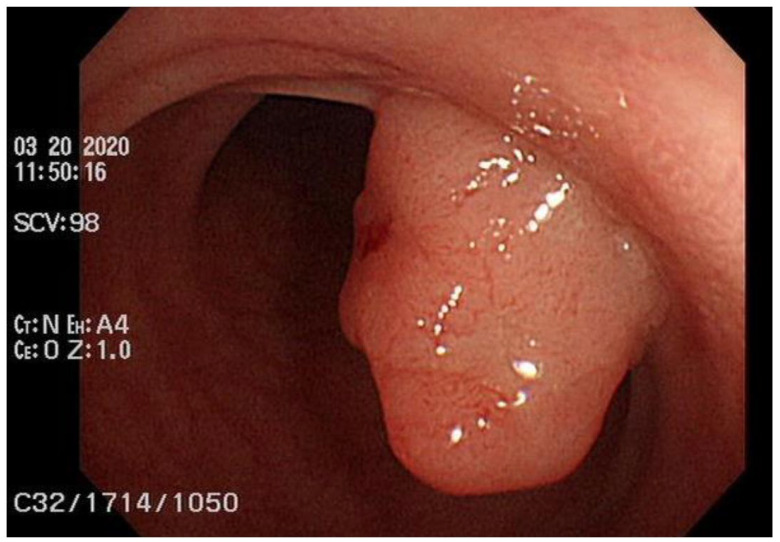
Colonoscopy with good bowel preparation. The scope reached the cecum smoothly and disclosed a 1.5 cm polyp in the sigmoid colon, which was proved to be adenocarcinoma after piecemeal endoscopic mucosal resection. The submitted specimen consisted of 11 tissue fragments measuring up to 1.5 × 1.3 × 0.9 cm in size. Histopathology report revealed colonic mucosa with the proliferation of atypical glandular cells arranged in irregular glands infiltrating in the desmoplastic stroma.

**Figure 3 diagnostics-13-02293-f003:**
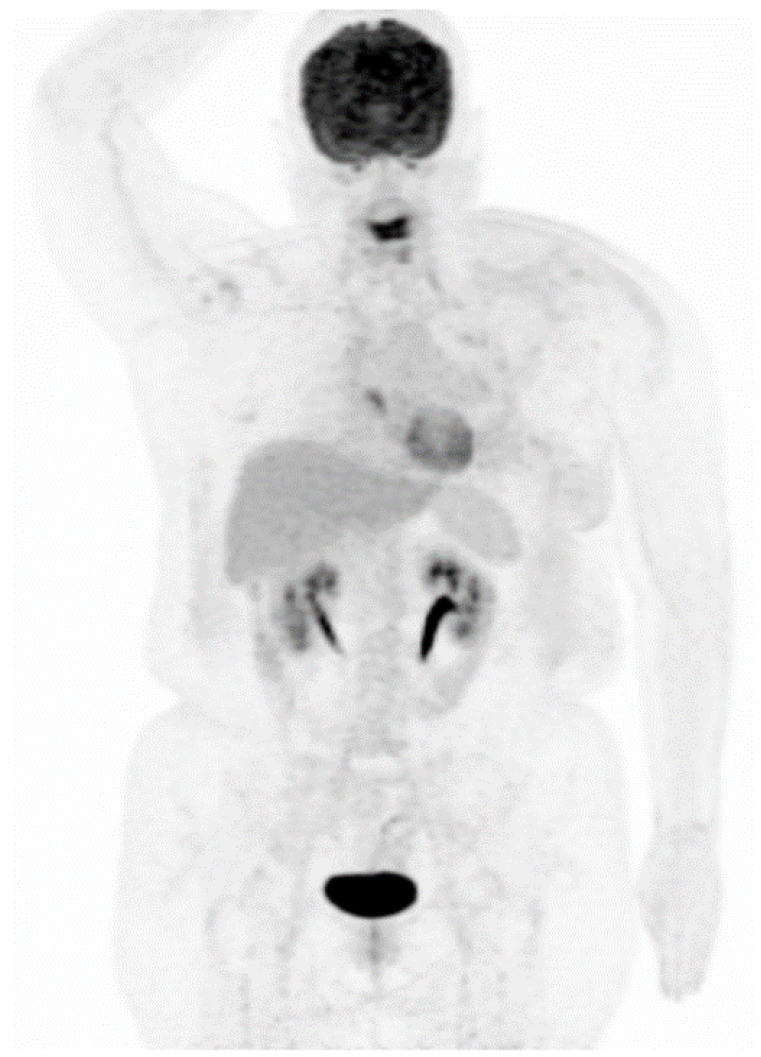
Maximum-intensity-projection image of FDG PET/CT. Follow-up FDG PET/CT scan 8 months later revealed an almost complete response of prior left breast cancer with multiple nodal, pulmonary, and bony metastases, with stationary status as compared with the second scan.

## Data Availability

Date available on request from the authors.

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
