# Peer review of "Utility of FDG PET/CT in Patient with Synchronous Breast and Colon Cancer"

_diagnostics, 2023, doi:10.3390/diagnostics13132293_

Round 1
Reviewer 1 Report
Dear authors,
The article presented a complex case with 2 synchronous cancers, metastatic invasive ductal carcinoma and colonic adenocarcinoma in a 60-year-old patient. This case report presents a particular complexity with multiple implications regarding diagnosis, treatment and genetic mechanisms.
The title is well chosen, the argument being given by the main imaging method that detected both pathologies and that was used for the follow-up.
It should be noted, the complexity of the case, in which the family history of breast cancer is mentioned, but unfortunately it would have been interesting for the patient to perform genetic testing.
The discussions are well argued, with clear differentiation between synchronous neoplasia and rare sites of breast cancer metastasis, detailing the use of PET-CT in the follow-up periods and the algorithm of treatment.
In conclusion, the current work makes an important contribution to the literature through the rarity of the presented case, of the well-diagnosed complex pathology and using a multidisciplinary treatment, bringing a plus to the clinicians who will have challenges like these in the future.
Kind regards
Author Response
Thank you for your reviewing our manuscript.

Reviewer 2 Report
This case report describes a case of a patient with breast and colon cancer diagnosed simultaneously.
The manuscript is Well written and of interest to the readers. However, it should be shortened and some of its parts should be reorganized.
Bellow please find a few specific comments and suggestions:
Title: the letter R is missing in "Cancer"
Introduction:
1. Suggest shortening the description of breast and colorectal cancer. These malignant diseases are well known.
2. Suggest reordering the paragraphs in the introduction: 1. a short description of breast cancer, 2. a short description of colorectal cancer, 3. a short definition of synchronous tumor.
3. The last paragraph in the introduction, starting with the words "cancer survey" and the explanation of FDG uptake mechanism, is unnecessary and can be deleted.
Case report section: no comments.
Figure 1 Legend: please shorten the description, and delete the technical details of the device, the acquisition protocol, and SUV calculations. The description of the image should start with the words: "Maximum intensity projection image of FDG PET/CT (A)...."
Discussion: No comments.
Author Response
- Title: the letter R is missing in "Cancer"
Answer: Corrected
- Introduction:
(1) Suggest shortening the description of breast and colorectal cancer. These malignant diseases are well known.
Answer: “Breast cancer can spread outside the breast through blood vessels and lymph vessels. When breast cancer spreads to other parts of the body, it is said to have metastasized” were deleted
(2) Suggest reordering the paragraphs in the introduction: 1. a short description of breast cancer, 2. a short description of colorectal cancer, 3. a short definition of synchronous tumor.
Answer: The paragraphs in the introduction were reordered according to the suggestion
(3) The last paragraph in the introduction, starting with the words "cancer survey" and the explanation of FDG uptake mechanism, is unnecessary and can be deleted.
Answer: “FDG is a glucose analog that is taken up by cellular glucose transport mechanisms and is phosphorylated by hexokinase. In most malignant cells, FDG-6-phosphate then becomes metabolically “trapped” intracellularly because of the relative lack of glucose-6-phosphatase activity in tumor cells.” were deleted
- Figure 1 Legend: please shorten the description, and delete the technical details of the device, the acquisition protocol, and SUV calculations. The description of the image should start with the words: "Maximum intensity projection image of FDG PET/CT (A)...."
Answer:
(1) The descriptions were rearranged according to the suggestion: “start with the words: Maximum intensity projection image of FDG PET/CT (A)....”
(2) Some descriptions of technical details and acquisition protocol were deleted.
(3) “The SUV was calculated as follows: SUV = (activity in ROI in mCi/mL)/(injected dose in mCi/patient's weight in kg).” were deleted.
